# Understanding the plume dynamics of explosive super-eruptions

Antonio Costa [1], Yujiro J. Suzuki[2] & Takehiro Koyaguchi[2]

Explosive super-eruptions can erupt up to thousands of $km^3$ of magma with extremely high mass flow rates (MFR). The plume dynamics of these super-eruptions are still poorly understood. To understand the processes operating in these plumes we used a fluid-dynamical model to simulate what happens at a range of MFR, from values generating intense Plinian columns, as did the 1991 Pinatubo eruption, to upper end-members resulting in co-ignimbrite plumes like Toba super-eruption. Here, we show that simple extrapolations of integral models for Plinian columns to those of super-eruption plumes are not valid and their dynamics diverge from current ideas of how volcanic plumes operate. The different regimes of air entrainment lead to different shaped plumes. For the upper end-members can generate local up-lifts above the main plume (over-plumes). These over-plumes can extend up to the mesosphere. Injecting volatiles into such heights would amplify their impact on Earth climate and ecosystems.

---

[1] Istituto Nazionale di Geofisica e Vulcanologia, Bologna 40128, Italy. [2] Earthquake Research Institute, University of Tokyo, 1-1-1 Yayoi, Bunkyo-ku, Tokyo 113-0032, Japan. Correspondence and requests for materials should be addressed to A.C. (email: antonio.costa@ingv.it)

Explosive super-eruptions eject from several hundreds to thousands of km$^3$ of magma at extremely high flow rates[1]. Many of these eruptions have had significant impacts to the climate and ecosystems[2–5]. Explosive super-eruptions cover areas within hundreds km from the vent with thick pyroclastic flows, blanket continent-size regions with ash, and inject large quantities of aerosols into the atmosphere[4]. Volatiles injected into the stratosphere can alter the Earth climate on a global scale even causing a volcanic winter that can persist for years to decades[2,3]. On the other hand, tephra layers associated with these catastrophic events are invaluable chronological markers across the affected regions[6,7]. The mass erupted during super-eruptions is orders of magnitude larger than the biggest eruptions experienced in historic times[1,4,8–10]. Estimates of mass flow rates (MFRs) during these super-eruptions, obtained from different independent approaches, suggest that they are extremely high, ranging from 10$^9$ to 10$^{11}$ kg/s[4,8,10–12]. Such large MFRs require multiple vents or continuous emission along dykes[13,14].

Plume dynamics of explosive super-eruptions are not well understood as such large events have not been witnessed. In order to understand how such volumes of material are ejected and dispersed we rely on field evidence and models that can produce the observed deposits. Our current understanding on how plumes of explosive super-eruptions behave is from extrapolations of simple integral models developed for describing columns generated from small MFR. These simple models[15–17] are based on the Buoyant Plume Theory (BPT) but the similarity assumption behind has been shown not to be valid for large MFRs[18].

Large explosive eruptions produce Plinian columns when the erupted mixtures of fragmented hot magma and gas entrain air, which heats up and expands making the plume buoyant. Above a critical MFR the eruption column becomes unstable[19] and collapses, producing pyroclastic flows that spreads laterally on the ground. At high MFR, the dilute parts of the hot pyroclastic flows can also become buoyant as they also entrain air, forming a co-ignimbrite eruption plume that can rise up to the stratosphere carrying massive quantities of elutriated fine ash and volatiles.

## Results

### Fluid-dynamical regimes of eruptive plumes for large MFRs.
Here, in order to avoid making unrealistic assumptions, we investigate the plume dynamics using a three-dimensional computational fluid-dynamical code (see Methods) designed to describe the evolution of volcanic plumes and umbrella clouds[19]. The code simulates the injection of a well-coupled mixture of solid pyroclasts (ash) and volcanic gas (assumed to be water vapour) from vents of different shapes above a flat surface into a stratified atmosphere. The model does not consider particle sedimentation and particle decoupling[20] but captures the plume dynamics (see Methods). In this study we will not consider the effects of the rotation of the Earth on the plume dynamics, which can be very significant for very large eruptions, affecting, among other things, their spreading and shape of the plume[21,22]. For this reason and the for the sake of simplicity, here we focus on eruptions occurring in the equatorial belt, where these effects are negligible[4,23] and consider tropical windless atmospheric conditions only (see Methods).

Considering the input parameters reported in Table 1 and atmospheric properties described in Methods, we explored the effects of variable MFRs for different vent geometries, such as a single circular vent, fissure, and vents at different distances. For the sake of simplicity, we focus on the results from a circular vent but these are rather general.

The fluid dynamics of large Plinian columns fed by a MFR~10$^9$ kg/s have been described in several studies[19,23,24]. In these columns, the fountain-like structure (radially suspended flow[24,25]) generated

**Table 1 Common input parameters and constants used for the three-dimensional simulations**

| Variable | Value |
|---|---|
| Exit velocity | 256 m/s |
| Exit temperature | 1053 K |
| Exit water fraction | 0.06 |
| Exit density | 3.5 kg/m$^3$ |
| Gas constant of volcanic gas | 462 J/kg/K |
| Gas constant of atmospheric air | 287 J/kg/K |
| Specific heat of solid pyroclasts | 1100 J/kg/K |
| Specific heat of volcanic gas at constant volume | 1348 J/kg/K |
| Specific heat of air at constant volume | 717 J/kg/K |
| Gravity body force | 9.81 m/s$^2$ |

in the lower part of the column is characterised by a high-concentration of erupted mixture and it is denser than the ambient air. In this region, the erupted mixture mixes with the air in large-scale vortexes and this mixture becomes rapidly buoyant (see Fig. 1a and Supplementary Movie 1). Approaching the critical MFR at ~10$^{9.5}$ kg/s (see Methods), the radially suspended flow[24,25] becomes unstable producing partial collapses, but the main plume still survives. The lower central part of the plume is a Negatively Buoyant Region (NBR), while the area around it is fed with relatively pure air that maintains its buoyancy and efficiently transports the mixture up to the stratosphere (see Fig. 1b and Supplementary Movie 2). The highest velocity remains in the central region generating a mushroom shape plume, with very high mass fractions in the central part up to the top of the plume (see Fig. 1b and Supplementary Movie 2).

Increasing MFR up to 10$^{10}$ kg/s produces a total collapse of the radially suspended flow, which generates continuous fountaining to the ground, feeding pyroclastic density currents and increasing the radius of the hotter NBR, resulting in a basal region (~60 km diameter) from where the large co-ignimbrite plume will develop (see Fig. 1c). In this case, because of the vigorous rising velocities in the periphery owing to the more effective local air entrainment, the upper central portion of such a large plume has a relatively low mass fraction compared with the outer region (Fig. 1c and Supplementary Movie 3). The resulting plume still maintains a mushroom shape but the plume top has flat, rather than umbonate, cap (Fig. 1c and Supplementary Movie 3).

A further increase of MFR up to 10$^{11}$ kg/s will produce a larger co-ignimbrite plume (>150 km in diameter). In this case, the vortices entrain air mainly at the periphery of the co-ignimbrite plume, without affecting the area around the hotter NBR (Fig. 1d and Supplementary Movies 4 and 5). This allows the periphery region of the co-ignimbrite plume to become much more buoyant and increase its velocity. Because of mass conservation, the vertical velocity in the inner part of the plume decreases. This regime results in the formation of a sort of toroid umbrella (donut-like shape), giving to the plume a depressed-cap mushroom shape (i.e., two separate lobes in a 2D cross-section).

Our simulations show that vent geometry has a strong control on the dynamics and stability of the Plinian columns but once co-ignimbrite plumes are generated the processes are predominantly controlled by the diameter of the co-ignimbrite plume, which is typically larger that the vent area, i.e., longer than the fissures or the distance between multiple vents (Supplementary Movies 1–6). For these reasons, the described features apply to all vent geometries despite the fact that a circular vent was used for the simulations (see Supplementary Movie 6).

### Implications for the assessment of eruption parameters.
MFRs are typically estimated from the total plume heights assuming BPT is valid. A similar approach has been extended to co-

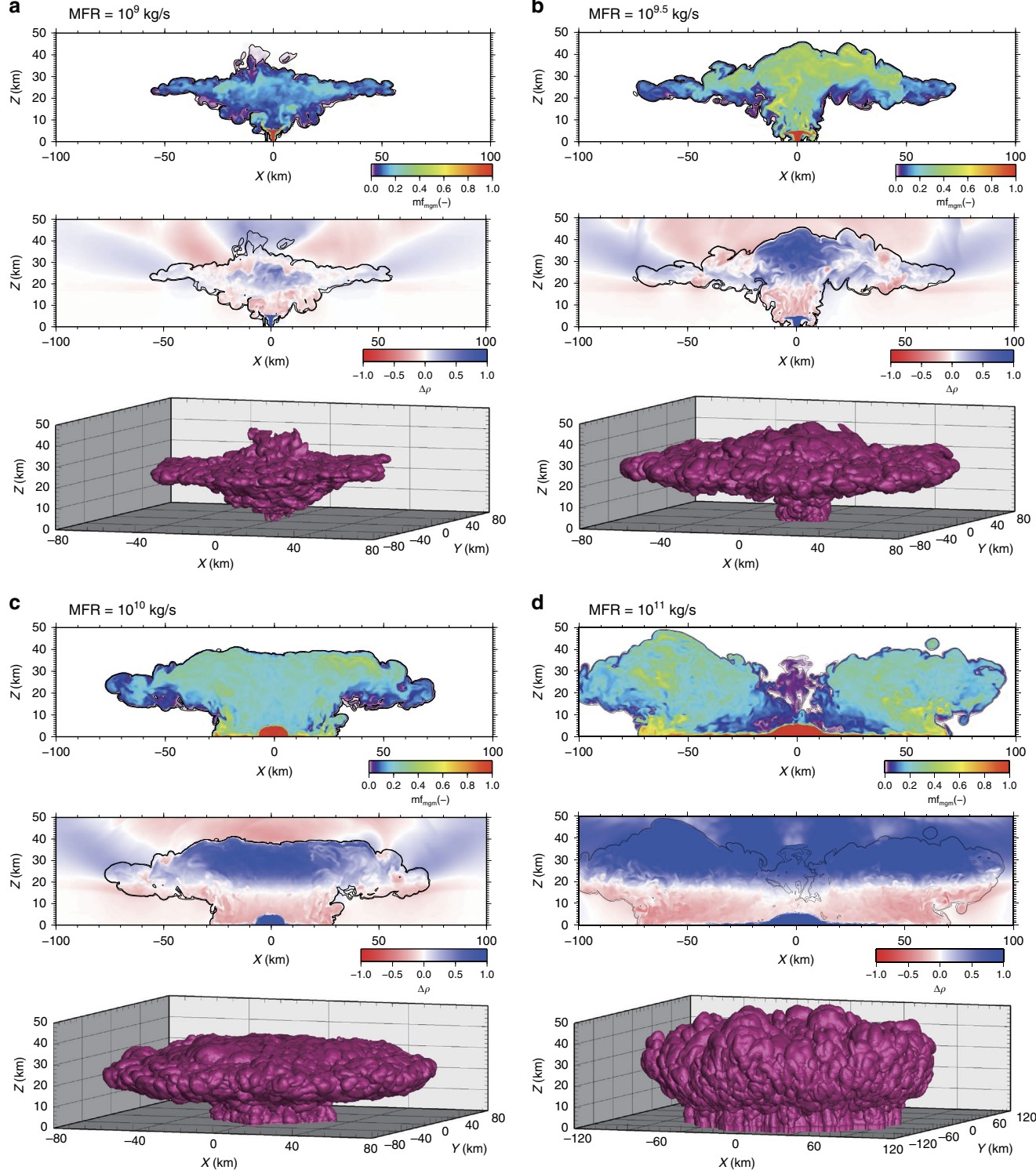

**Fig. 1** Simulation results for the plumes with different MFRs. **a** $10^9$ kg/s, **b** $10^{9.5}$ kg/s, **c** $10^{10}$ kg/s, **d** $10^{11}$ kg/s. The figures show snapshots at $t = 800$ s after the eruption initiation: vertical cross-sections of the mass fraction of the erupted mixture is 0.01 (**a**–**d**, upper panels); density difference relative to the atmospheric density (**a**–**d**, middle panels); three-dimensional isosurface where the mass fraction of the erupted mixture is 0.01 (**a**–**d**, lower panels)

ignimbrite plume[15] and is still largely used in the volcanological community[16,17]. However, the dynamics of large co-ignimbrite plumes are markedly different from BPT as their horizontal extension is typically much larger than their height. The simulations indicate that, above a critical MFR, the co-ignimbrite plume rises from a source with radius $R_{CI}$, increasing with MFR as by $R_{CI} \approx 2.8 \cdot 10^{-4}\sqrt{\text{MFR}}$ (with the co-ignimbrite plume radius, $R_{CI}$, expressed in km and MFR in kg/s, see Fig. 2); this

power-law dependence of the MFR with run-out distance was predicted in simple models of pyroclastic flows[26]. The mechanisms of air entrainment from such broad sources are profoundly different from, and invalidate the similarity assumption used in, BPT. The difference in the scale of the horizontal extension of plume also affects the behaviour of the upper part of the plume, including the column height and the dynamics in the umbrella cloud.

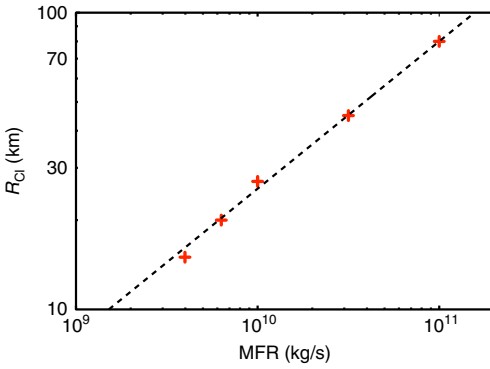

**Fig. 2** Co-ignimbrite plume radius as function of MFR. Cross denotes values estimated from the 3D simulations and the line the empirical relationship given by $R_{CI} \approx a \cdot MFR^b$ with $a \cong 2.8 \cdot 10^{-4} \, km \, kg^{-b} \, s^b$ and $b \cong 0.5$. Note that the model simulates a dusty-gas mixture, whereas the pyroclastic flows from such super-eruptions can be formed of concentrated dispersions, which depend on imperfect coupling between gas and particles[10]

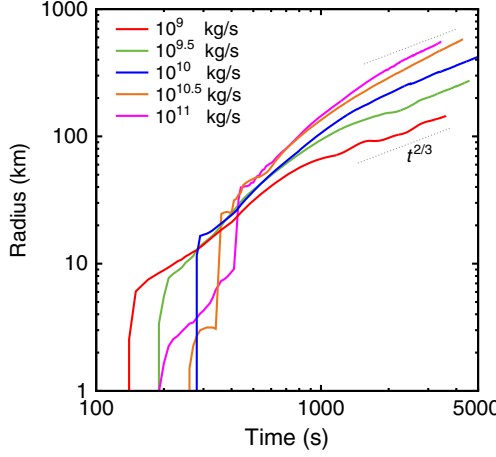

**Fig. 4** Radii of the spreading umbrella clouds as a function of the time for the different simulations. Dashed lines correspond to the relationships between radius, $R$, and time, $t$: $R(t) = At^{2/3}$, where $A$ is the proportionality constant. This dependence is commonly used to estimate MFR[23,35,36]. The plot shows that with respect to the Plinian case, co-ignimbrite plumes with MFR $> 10^{10}$ kg/s have a steeper spreading rate, with the power-law exponent[23,35,36] passing from ~0.66 to 0.96. These results validate the robustness of simple box models for describing the umbrella region[35,36]. However, the fact that the spreading of the umbrella cloud with time is faster for co-ignimbrite plumes with respect Plinian plumes suggests caution when we use relationships obtained for the latter to estimate MFR above the critical value

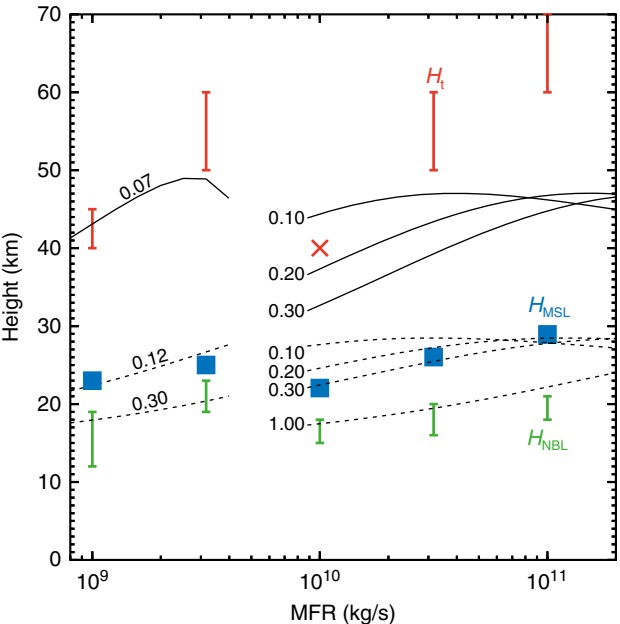

**Fig. 3** Characteristic plume heights as a function of the MFR. Red bars represent the height over the whole plume ($H_t$). Blue squares represent the altitude of maximum level of radial spreading of the umbrella cloud ($H_{MSL}$). Green bars represent the altitude of NBL ($H_{NBL}$). The characteristic heights from the BPT model are shown as dashed and solid curves. In this diagram, the total heights are compared with the heights where the upward velocity becomes zero in the BPT calculations, whereas the MSLs and NBLs are compared with the NBL calculated by the BPT model (see also Table 2). Values on the curves are the effective entrainment coefficients that explain the 3D results. In the calculations of co-ignimbrite ash clouds (the MFR > $10^{10}$ kg/s) using the BPT model, the initial temperature and mass fraction of the entrained air were estimated so that the initial density is equal to the atmospheric density near the ground (see Table 3). According to this mass fraction, the effective MFR at the ground can be estimated. The initial radii were calculated using the scaling law reported in Fig. 2

Here, we focus on the dependence of maximum plume height with MFR (Fig. 3 and Fig. 4 for the dynamics of umbrella cloud). The result shows that from $10^9$ to $10^{10}$ kg/s the maximum plume heights remain similar and are between 40 and 60 km. Co-ignimbrite plumes appear steadier than Plinian columns, which show a more oscillatory behaviour (see Supplementary Movies 1–5).

The maximum column height and highest mass fraction in the umbrella region is reached for MFR around the critical value, i.e., $10^{9.5}$ kg/s. Remarkably, the Neutral Buoyancy Level (NBL) remains at ~20 km for all the simulations with MFRs above the critical MFR (see Table 2).

For MFR of ~$10^{11}$ kg/s, maximum plume height is in the peripheral region rather than in the centre, owing to the more efficient entrainment of air from the border of the plume. The maximum height for the bulk mass is at ~50 km but local up-lifts, having a diameter of ~30–40 km, develop above the umbrella region (see Supplementary Figure 1) and keep rising up to the mesosphere (60–70 km). In this case, two different effective heights should be considered, one for the bulk mass spreading around the umbrella region and one for the maximum height reached by the local up-lifts (we call them "local over-plume" hereafter). These local over-plumes develop from the base of the periphery of the co-ignimbrite plumes, because of local heterogeneities in the efficiency of air entrainment, and are characterised by higher velocities and larger mass fractions (see Supplementary Figure 1).

The complex relationship between plume height and MFR in Fig. 3 suggests that for large eruption intensities (MFR > $10^9$ kg/s) we cannot use column height estimations to assess the value of MFR[15,27]. To compare our results from the three-dimensional simulations with those of the simple BPT integral models[15,27], we estimated the main mean variables[20], such as mixture density difference, $\Delta \rho$, vertical velocity, $U$, temperature, $T$, and mass fraction $\xi$ (see Supplementary Figure 2) and extracted optimal parameter values (see Fig. 3 and Table 3). These BPT models do not adequately describe co-ignimbrite plumes but if they are used as extrapolations the effective entrainment coefficient, $k$, should be properly tuned and not assumed as an invariant. Since variations of $H_{NBL}$ with MFR from sustained Plinian column to fully co-ignimbrite plume are almost negligible ($H_{NBL}$ ~15–20 km, see Table 2), accordingly to BPT, this implies that the effective entrainment coefficient should increase with MFR and have significantly different values for the two regimes.

**Table 2 Typical plume heights for different MFRs**

| Plume dynamics | | | MFR (kg/s) | | |
|---|---|---|---|---|---|
| | $10^9$ | $10^{9.5}$ | $10^{10}$ | $10^{10.5}$ | $10^{11}$ |
| $H_{NBL}$ (km)[a] | 12–19 | 19–23 | 15–18 | 16–20 | 18–21 |
| $H_{MSL}$ (km)[b] | 23 | 25 | 22 | 26 | 29 |
| $H_t$ (km)[c] | 40–45 | 50–60 | 40 | 50–60 | 60–70 |
| Mean $\xi$ in the umbrella | Low | Very high | Medium | High | Very high |

[a] The lower value corresponds to the column region, while the upper value to the entire umbrella region
[b] The radial maximum spreading level was estimated as the height containing the greatest mass of airborne particles[34]
[c] The lower value corresponds to dilute part of the plume ($\xi = 10^{-3}$), while the upper value to the concentrated part of the plume ($\xi = 10^{-1}$)

**Table 3 Input parameters for BPT model inferred from three-dimensional simulations**

| Parameter | MFR (kg/s) | | | | |
|---|---|---|---|---|---|
| | $10^9$ | $10^{9.5}$ | $10^{10}$ | $10^{10.5}$ | $10^{11}$ |
| Plume radius (km) | 0.60 | 1.06 | 28.0 | 50.0 | 88.5 |
| Exit velocity (m/s) | 256 | 256 | 4.87 | 4.87 | 4.87 |
| Exit temperature (K) | 1053 | 1053 | 858 | 858 | 858 |
| Exit gas fraction | 0.06 | 0.06 | 0.33 | 0.33 | 0.33 |

The total column height, $H_t$, varies with MFR (see Table 2), rising from $H_t \cong 40 - 45$ km at MFR $= 10^9$ kg/s to $H_t \cong 50 - 60$ km at MFR $= 10^{9.5}$ kg/s, then decreasing $H_t \cong 40$ km at MFR $= 10^{10}$ kg/s and increasing again to $H_t \cong 50 - 60$ km at MFR $= 10^{10.5}$ kg/s. To estimate plume height ranges, we considered a fraction mass of $\xi = 10^{-3}$ for the dilute upper region and $\xi = 10^{-1}$ for the lower values, which refer to a more concentrated plume region. This implies that the behaviour of the plumes above NBL and the plume height is controlled by an effective entrainment coefficient, $k_U$, which is different from the entrainment coefficient, $k_L$, that governs the air entrainment below NBL (see Fig. 3). The total heights obtained from three-dimensional simulations cannot be described by the BPT model (for the optimal input values see Table 3) even if the value of $k_U$ is empirically tuned. The maximum height is reached by the oscillating local over-plume at MFR $> 10^{10.5}$ kg/s. It is inferred that the total column height (reported in Fig. 3) is generated from a large amplitude gravity wave that is excited by the intensive plume. This again confirms the difficulty of simple integral models to capture such complex plume dynamics.

## Discussion

Our results have enormous implications for the assessment of the dynamics of super-eruptions. For the most extreme MFRs[13], $\sim10^{11}$ kg/s, similar to those estimated for the Young Toba Tuff (YTT) eruption[4], the total plume height increases beyond the stratosphere (up to $\sim$60–70 km) owing to the development of local over-plumes above the umbrella region, even though the NBL remains at $\sim$20 km. The bulk mass spreads in the umbrella region between $\sim$20 and $\sim$50 km, and the local over-plumes develop above the main umbrella region (see Supplementary Figure 1). All previous studies relied on the results of the Woods and Wohletz model[15] and used a relatively low plume height ($\sim$30–40 km) but our simulations indicate that volatiles and fine ash can be transported up to 70 km. This has important implications for the effects of eruptions, like YTT, on the Earth's climate[2,3,28]. Climate models are sensitive to the injection height because atmosphere stratification and availability of $H_2O$ influence the conversion of $SO_2$ into sulphate, which governs the climate response. Injections into the stratosphere affect the albedo of the atmosphere on the order of decades[2] but the longevity of $SO_2$ in the mesosphere could be considerably longer.

Our simulations also show that partially collapsing plumes, generated slightly above the critical MFR value, can reach heights of up to 50–60 km and efficiently transport the mixture up to the stratosphere (see Fig. 1b and Supplementary Movie 2). This results in effective upward transport of a large mass fraction of fine ash that is generated from the pyroclastic flows, enriching the fine ash content of the umbrella cloud with respect to the ground-hugging pyroclastic flows. This can be the case for eruptions similar to the Campanian Ignimbrite event for which a MFR of $\sim10^9$ kg/s was empirically estimated for the initial Plinian phase and $\sim2$–$5\times10^9$ kg/s for the co-ignimbrite phase[12,29].

## Methods

**Computational fluid-dynamical model.** To simulate fluid dynamics of volcanic plumes, we used the pseudo gas model by Suzuki et al.[19], in which the momentum and heat exchanges between the volcanic ash and gas are assumed to be so rapid that the velocity and temperature are the same for all phases. The fluid dynamics model solves a set of partial differential equations describing the conservation of mass, momentum, and energy, and a set of constitutive equations describing the thermodynamics state of the mixture of volcanic ash, gas, and air. The computational model was designed to reproduce the injection of a mixture of volcanic ash and volcanic gas from a vent into a stratified atmosphere. For this study typical tropical atmosphere conditions were used as initial atmospheric conditions (see Supplementary Figure 3). For given temperature gradients of atmosphere, the atmospheric density and pressure were calculated from the hydrostatic relationship. Initial wind velocity was set to be zero in the whole computational domain. Free slip conditions were applied at the ground boundary, and fixed inflow conditions at the vent. At the other boundaries, mass, momentum, and energy fluxes were assumed to be continuous (i.e., free outflow/inflow conditions). The use of the latter conditions can avoid the wave reflection at the boundaries even when the initial atmospheric conditions, such as hydrostatic pressure and density, are perturbed by the eruption.

The governing equations were solved numerically by the Roe scheme[30] with MUSCL (Monotone Upstream-centred Scheme for Conservation Laws) interpolation[31] for spatial integration and time splitting method for time integration. The present model with these schemes has a third-order accuracy in space and second-order accuracy in time. We used a generalised coordinate system in the computational domain.

**Computational settings and simulation strategy.** The grid size near the vent was set to be sufficiently smaller than the vent diameter (i.e., $D_0/20$, where $D_0$ is the vent diameter) in order to resolve the flow structures and turbulent mixing at a low altitude, whereas it was increased at a constant rate up to 300 m/grid with the distance from the vent. We confirmed that the flow regime and plume height are basically unaffected by the numerical resolutions by changing the grid sizes (see Supplementary Figure 4).

The computations were carried out on the Earth Simulator (NEC SX-ACE; 64 GFLOPS/core) at the JAMSTEC and also on the Fujitsu PRIMERGY CX400 (22 GFLOPS/core) at the Research Institute, Kyushu University. Each simulation took 52–280 h with 512 cores.

In order to investigate how variation of MFR affects the plume dynamics, we carried out a set of simulations with variable vent size, following the method described by Suzuki et al.[25]. The vent radius ranged from 600 to 6000 m. The pressure, density, and velocity at the vent were kept fixed among the simulations. The pressure at the vent was assumed to be in equilibrium with the atmospheric pressure. We assumed a magmatic temperature of 1053 K and water content of 6 wt.%, similar to the magmatic properties in the Pinatubo 1991 eruption[23]. Considering these values of pressure, temperature, and density the equation of state gives a density at the vent of 3.5 kg/m³. The exit velocity was assumed to be 256 m/s, corresponding to a fixed Mach number of 1.5. As a result, the MFR ranges from $10^9$ to $10^{11}$ kg/s.

It is important to stress that in this study we kept fixed the magmatic properties such as magma temperature and water content. The variation of these properties

can lead to different critical conditions between the flow regimes described in the main text. However, the qualitative features of each flow regimes would be same in the possible ranges of magmatic temperature and water content for magmatic eruptions. We also assumed the equilibrium of pressure and the supersonic flow as the conditions at the vent. The disequilibrium and sub/supersonic flow can change the flow structures near the vent, which results in the change of the final distance of PDC and therefore the change of transition between the flow regimes[32,33]. In addition, we ignore the non-equilibrium effects between the volcanic ash and gas phases. However, the non-equilibrium effects are less relevant in the strong eruptions rather than in the weak eruptions[20].

**Data availability**. The authors declare that all data supporting the findings of this study are available in the article and in Supplementary Information. Additional information is available from the corresponding author upon request.

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

## Acknowledgements

A.C. was partially supported by a grant of the International Research Promotion Office Earthquake Research Institute, the University of Tokyo. Y.J.S. and T.K. were partially supported by KAKENHI (grand nos. 25750142 and 17K01323). We warmly thank V.C. Smith for revising the English and very helpful suggestions.

## Author contributions

A.C. and Y.J.S. designed the simulation set. Y.J.S. performed the runs and processed the results. A.C., Y.J.S., and T.K. analysed and interpreted the results. A.C. wrote the manuscript with input from all the co-authors.

## Additional information

**Competing interests:** The authors declare no competing financial interests.

