## [Peer Review File · Nature Communications]

Reviewers' Comments:

Reviewer #1:

Remarks to the Author:

This paper aims to provide a dynamical description of explosive super-eruptions. On the whole, the material is well-presented, with good graphics and details of how the mathematical model is constructed. However, I cannot recommend its acceptance in its present form because it omits a main feature of the dynamics of very large eruptions, namely the rotation of the earth. This will have a significant effect on the flows that they describe, affecting, among other things, their spreading and shape.

The relevant effects are described in two papers (which the authors do not reference) as follows:

Baines, P. G., and R. S. J. Sparks (2005), Dynamics of giant volcanic ash clouds from supervolcanic eruptions, *Geophys. Res. Lett.*, 32, L24808, doi:10.1029/2005GL024597,

BAINES, P.G. JONES, M.J. and SPARKS S.J. 2008: The variation of large-magnitude volcanic ash cloud formation with source latitude. *J. Geophys. Res.*, 113, D21204, doi: 10.1029/2007JD009568.

I have minor quibbles with details of the paper, but since the objective of the authors is describe VERY LARGE eruptions, there is little point in listing them here.

Reviewer #2:

None

Reviewer #3:

Remarks to the Author:

Review report for Costa et al.

G.A. Valentine

I found this to be an excellent paper that is definitely worthy of publication in *Nature*

Communications, and it follows other papers in this journal on the topic of super-eruptions. It is very well written, cutting edge work that will be of general interest (who isn't interested in super-eruptions?!).

Major claims/findings – The authors for the first time present 3-D simulations of super-eruptions. They demonstrate the unusual geometry of the eruption clouds, which negates fundamental assumptions that are built into most volcanic plume models. The authors demonstrate and analyze the resulting differences. The paper shows that small plumes that grown from the main cloud can reach tremendous heights in the atmosphere – to my knowledge this is the first result that strongly suggests volcanic plumes reaching more than 50 km altitude. This paper forms a foundation for follow-on studies of potential regional and global impacts of such eruptions.

Are the findings novel and original? – Yes, the findings are novel and the application of the 3-D numerical model is cutting edge. To my knowledge the authors are the first ones to really tackle, using modern computational fluid dynamics tools, this difficult but intriguing problem of super-eruption dynamics.

Are the results of interest to others in the community and to natural sciences generally? – Yes, super-eruptions are compelling from a broad earth and atmospheric sciences

perspective, but are also of interest to natural scientists in general.

This paper will have a strong influence on the field.

I made several minor comments in the manuscript, mainly some editorial suggestions, but also some issues that need to be – and can easily be – mentioned or addressed. First, in the discussion of runout distance (of pyroclastic density currents), the authors need to have some brief statement that the PDC dynamics (which determine their runout; see Roche et al. 2016, Nature Communications) are not fully captured in their model (or in the simple models they refer to). Again, this does not negate the work, it's just a caveat that needs to be made. Also, in the methods section, there could be more information on the computations especially on the mesh/grid and how the initial and boundary conditions were defined (they mention outflow boundaries, for example, but are these held at a constant P-T as defined by a standard atmospheric profile?). The mesh/grid should be justified by a sensitivity analysis, or some rationale given for why it is appropriate.

Again, these issues can be easily addressed and I look forward (hope) to see the paper published soon in Nature Communications.

Response to the reviewers

Reviewers' comments:

Reviewer #1 (Remarks to the Author):

This paper aims to provide a dynamical description of explosive super-eruptions. On the whole, the material is well-presented, with good graphics and details of how the mathematical model is constructed. However, I cannot recommend its acceptance in its present form because it omits a main feature of the dynamics of very large eruptions, namely the rotation of the earth. This will have a significant effect on the flows that they describe, affecting, among other things, their spreading and shape.

The relevant effects are described in two papers (which the authors do not reference) as follows:

Baines, P. G., and R. S. J. Sparks (2005), Dynamics of giant volcanic ash clouds from supervolcanic eruptions, *Geophys. Res. Lett.*, 32, L24808, doi:10.1029/2005GL024597,
BAINES, P.G. JONES, M.J. and SPARKS S.J. 2008: The variation of large-magnitude volcanic ash cloud formation with source latitude. *J. Geophys. Res.*, 113, D21204, doi:10.1029/2007JD009568.

I have minor quibbles with details of the paper, but since the objective of the authors is describe VERY LARGE eruptions, there is little point in listing them here.

We are grateful to the Reviewer#1 for highlighting this point. We knew that the effects of Earth rotation can be significant and the results of the suggested articles. However, because the manuscript was originally submitted to Nature and, from there directly transferred to Nature Communications, there was a limitation on the number of references. Moreover, because our manuscript is the first study presenting full plume dynamics associated to VERY LARGE eruptions, we think that it is better and clearer not considering the effects due to Coriolis force, which will be studied in other works in the near future. For similar reasons we considered a windless atmosphere as we know the effect are not significant in this regime (Costa et al., 2016).

However, in the revised version we explicitly discussed this limitation and we justify it because for this study we focus on eruptions occurring near the Equator and, for this reason, we added an Extended Figure showing the profile we used that are typical of a tropical atmosphere and the following sentence at lines 68-72:

In this study we will not consider the effects of the rotation of the Earth on the plume dynamics, which can be very significant for very large eruptions, affecting, among other things, their spreading and shape of the plume^{20,21}. For this reason and the for the sake of simplicity, here we focus on eruptions occurring in the equatorial belt,

where these effects are negligible^{4,23} and consider tropical windless atmospheric conditions only (see Methods).

Reviewer #2 (Remarks to the Author): See attached report
Reviewer #3 (Remarks to the Author):

General comments.

This article is a direct continuation of the paper. Costa, A., et al.; Results of the eruption column model inter-comparison study, J. Volcanol. Geotherm. Res., 326, 2-25, doi:10.1016/j.jvolgeores.2016.01.017 (2016), reference no 18 in the manuscript. The subject is quite interesting, estimation of the output of very large eruption events.

Such an estimate would give a relation between the column height and the mass flow rate of the eruption. The uncertainty in this estimate is one of the major obstacles in predicting the correct size of volcanic clouds. Too large predicted clouds were an important cause in the 2010 Eyjafjallajökull disaster to name a well-known example of the importance of reliable models.

The relations of Sparks and Mastin, results in the famous relation: Mass flow is proportional to column height in power 3 – 4. This model and similar models suffer from the classical flaw not containing all the nondimensional parameters that dimensional analysis known from fluid mechanics would result in. The various models in ref 18 are of this type. One nondimensional parameter a full dimensional analysis would bring about is the R/H, radius of the vent/height of column, this parameter is not included in conventional eruption models so the effect of it is unclear.

It so happens that the calculations in the paper use fixed exit velocity V and density ρ listed in Table 1. Now ejected mass flow $M = \rho VA = \rho V \pi R^2$ so variation of M from $10^9 - 10^{11}$ means multiplication of the initial R1 by 10 and no other change in the output parameters. According to the old models $M/M_0 = (H/H_0)^4$ which can be expected to hold within a factor 10. This seems to be the case in Figure 3 as $M/M_0 = 100$ gives $H/H_0 =$ about 3.

The following improvements are suggested.

The authors should include the mathematics of their fluid dynamical system, enter the fixed initial values and show the remaining nondimensional parameters and discuss their importance. A discussion of the effect of inhomogeneous atmospheric conditions (inversions) have on the plume spreading is also welcome.

When this is done the article can be reconsidered for publication.

Specific comments

Considering the rewriting suggested above makes specific comments unnecessary at this stage.

We thank the Reviewer#2 for her/his comments. Concerning the dimensionless variables, H/R can be used to characterize the regime only for the sustained eruptive columns (i.e. below the critical MFR) but it is not representative of co-

ignimbrite plumes where the radius of the plume (scaling with MFR as shown in Fig. 2) is a crucial variable and can be even larger than the plume height itself (a more detailed study on the scaling is the subject of ongoing research).

Concerning the suggested improvements we added a better description of the fluid dynamical model, improving the explanation of the adopted initial and boundary conditions (see lines 319-339). Effects of atmospheric stratification were already included as we considered meteorological profiles typical of tropical region. In the revised version we explicitly added the used profiles as a supplementary figure (Extended Data Figure 2). However, similarly to the plume of very large Plinian eruptions, the effects of atmosphere stratification are very important. The latter point was already discussed in Suzuki et al. (2005), Suzuki and Koyaguchi (2009) and other references (e.g. Ishimine, Y. (2007), A simple integral model of buoyancy-generating plumes and its application to volcanic eruption columns, J. Geophys. Res., 112, B03210, doi:10.1029/2006JB004274). The relationship between typical plume heights and MFR is however reported in Figure 3.

Finally, in order to improve the readability and furnish the values of the dimensions of the simulated plumes, since in Nature Communications we have more space, we moved the Extended Table 1 and 2 in the main text. In this way the reader can easily estimate the dimensionless scaling (see new Table 2 for plume heights and Table 3 for plume radius).

Review report for Costa et al. G.A. Valentine: I found this to be an excellent paper that is definitely worthy of publication in Nature Communications, and it follows other papers in this journal on the topic of super-eruptions. It is very well written, cutting edge work that will be of general interest (who isn't interested in super-eruptions?!).

Major claims/findings – The authors for the first time present 3-D simulations of super-eruptions. They demonstrate the unusual geometry of the eruption clouds, which negates fundamental assumptions that are built into most volcanic plume models. The authors demonstrate and analyze the resulting differences. The paper shows that small plumes that grown from the main cloud can reach tremendous heights in the atmosphere – to my knowledge this is the first result that strongly suggests volcanic plumes reaching more than 50 km altitude. This paper forms a foundation for follow-on studies of potential regional and global impacts of such eruptions.

Are the findings novel and original? – Yes, the findings are novel and the application of the 3-D numerical model is cutting edge. To my knowledge the authors are the first ones to really tackle, using modern computational fluid dynamics tools, this difficult but intriguing problem of super-eruption dynamics.

Are the results of interest to others in the community and to natural sciences generally? – Yes, super-eruptions are compelling from a broad earth and atmospheric sciences perspective, but are also of interest to natural scientists in general.

This paper will have a strong influence on the field. I made several minor comments in the manuscript, mainly some editorial suggestions, but also some issues that need to be – and can easily be – mentioned or addressed. First, in the

discussion of runout distance (of pyroclastic density currents), the authors need to have some brief statement that the PDC dynamics (which determine their runout; see Roche et al. 2016, Nature Communications) are not fully captured in their model (or in the simple models they refer to). Again, this does not negate the work, it's just a caveat that needs to be made. Also, in the methods section, there could be more information on the computations especially on the mesh/grid and how the initial and boundary conditions were defined (they mention outflow boundaries, for example, but are these held at a constant P-T as defined by a standard atmospheric profile?). The mesh/grid should be justified by a sensitivity analysis, or some rationale given for why it is appropriate. Again, these issues can be easily addressed and I look forward (hope) to see the paper published soon in Nature Communications.

We are deeply grateful to the Reviewer#3 for his comments and suggestions. In the revised version we incorporated all his suggestions and improved that critical points.

In particular:

- we explicitly added a sentence explaining the limitations of the model in the caption of Fig. 2:

Note that the model simulates a dusty-gas mixture, whereas the pyroclastic flows from such super-eruptions can be formed of concentrated dispersions, which depend on imperfect coupling between gas and particles¹⁰.

- we improved the description of the initial/boundary conditions used for the model (see lines 319-339) and reported some results of a sensitivity study on the computational grid size as Extended Data (Extended Data Figure 4);

- we made all the corrections suggested on the annotated manuscript.